# Effect of Stimulus Size in a Visual ERP-Based BCI under RSVP

**DOI:** 10.3390/s22239505

**Published:** 2022-12-05

**Authors:** Álvaro Fernández-Rodríguez, Aube Darves-Bornoz, Francisco Velasco-Álvarez, Ricardo Ron-Angevin

**Affiliations:** Departamento de Tecnología Electrónica, Universidad de Málaga, 29071 Malaga, Spain

**Keywords:** brain–computer interface (BCI), event-related potential (ERP), rapid serial visual presentation (RSVP), stimulus, size

## Abstract

Rapid serial visual presentation (RSVP) is currently one of the most suitable paradigms for use with a visual brain–computer interface based on event-related potentials (ERP-BCI) by patients with a lack of ocular motility. However, gaze-independent paradigms have not been studied as closely as gaze-dependent ones, and variables such as the sizes of the stimuli presented have not yet been explored under RSVP. Hence, the aim of the present work is to assess whether stimulus size has an impact on ERP-BCI performance under the RSVP paradigm. Twelve participants tested the ERP-BCI under RSVP using three different stimulus sizes: small (0.1 × 0.1 cm), medium (1.9 × 1.8 cm), and large (20.05 × 19.9 cm) at 60 cm. The results showed significant differences in accuracy between the conditions; the larger the stimulus, the better the accuracy obtained. It was also shown that these differences were not due to incorrect perception of the stimuli since there was no effect from the size in a perceptual discrimination task. The present work therefore shows that stimulus size has an impact on the performance of an ERP-BCI under RSVP. This finding should be considered by future ERP-BCI proposals aimed at users who need gaze-independent systems.

## 1. Introduction

A brain–computer interface (BCI) is a type of assistive technology (AT) that allows a user to communicate with his/her environment using only brain signals [1]. One of the purposes of AT is to facilitate the user’s interaction with his/her environment [2]. In addition to BCI, there are several types of ATs, such as eye trackers, head-pointing devices, or low-pressure sensors. However, some injuries or diseases, such as amyotrophic lateral sclerosis (ALS), can lead to situations in which the muscular channel and even eye movements can be affected [3]. Therefore, in severe motor limitations, most of these examples of AT may no longer be useful because they depend on some type of muscular channel that may be affected in the patient. This makes BCIs a promising option in severe cases of lack of muscular control

Although BCI systems have used various types of input signals, electroencephalographic (EEG) signals are currently the most widely used due to their portability, relatively low cost, and adequate temporal resolution, among other reasons [4]. Some of the most commonly used EEG signals are sensorimotor rhythms (SMRs), steady-state visual evoked potentials (SSVEPs), code-modulated visual evoked potentials (c-VEPs), and event-related potentials (ERPs). Of these, ERPs are the most widely used to control specific paradigms that do not require ocular mobility. This is due to the fact that, for example, in the case of spellers (a specific tool for selecting letters on an interface, which allows for verbal communication), they usually allow a large number of commands and do not require extensive training to yield adequate accuracy [5]. It therefore seems appropriate to employ ERPs as a brain signal for the control of a BCI system.

ERPs are changes in the voltage of the electrical activity of the brain produced by the presentation of a specific event. These events may be external stimuli presented via different modalities, such as visual, auditory, or tactile events [6]. The modality used in the present work is the visual one. According to a review by Allison et al. [6], this modality generally provides the best results for the control of a BCI based on ERPs (ERP-BCI). In addition, under certain presentation paradigms, the visual modality can be employed even if the user does not have ocular control. One paradigm that does not require ocular mobility is a rapid serial visual presentation (RSVP) (e.g., Acqualagna and Blankertz [7]). In the following text, we explain how RSVP is used for control of a visual ERP-BCI.

The main feature of RSVP is that the visual stimuli are presented serially (one after the other) in the same spatial location. For the control of a visual ERP-BCI, different visual stimuli are presented to the user, who must attend to one of them. Paying attention to the desired stimulus (for example, a letter in the case of a speller) should elicit a different electrical signal in the brain than the signal associated with undesired stimuli. Hence, the objective of an ERP-BCI is to discriminate between the desired or attended stimulus (target) and undesired or non-attended stimuli (non-target) based on the user’s brain signals. The main component used by these systems is P3 (also called P300). This is a positive deflection in the amplitude of the brain’s electrical signal that begins approximately 300–600 ms after the presentation of a stimulus that the user is expecting (target). However, an ERP-BCI generally uses all possible ERPs involved in the observed time interval (e.g., P2, N2, or a late positive potential). That is, any signal that helps to discriminate the attended stimulus (target) from unattended ones (non-target) will be used in the selected interval time (e.g., 0–800 ms).

As mentioned above, the target population for a visual ERP-BCI may be patients who have lost even the ability to control their eyes. It is therefore important that the interfaces offered to this type of user are adapted to their abilities. For example, performance worsens considerably if the user cannot directly attend to stimuli with the gaze [8,9]. This makes it convenient to employ paradigms that do not require eye control to yield adequate performance, such as RSVP. Other works have shown that parameters such as (i) the type of stimulus employed [10], (ii) the stimulus duration [11], and (iii) the spatial distribution of the stimuli [12] have an impact on performance.

Stimulus size is a relevant factor because other characteristics may depend on it, such as the size of the screen or the distance at which the user has to attend to the stimuli. To our knowledge, the effect of stimulus size in an ERP-BCI under RSVP (i.e., with the stimuli presented in the same position and completely overlapped) has not previously been studied. Using other paradigms in addition to RSVP, several studies have explored whether the stimulus size can affect performance in an ERP-BCI. However, as will be shown below, these studies have not provided a conclusive answer. Li et al. [13] and Ron-Angevin et al. [14] published two papers that presented the stimuli in the form of a matrix via a row-column flash pattern and reported an effect due to the size of the interface employed (including the size of the stimuli and the distance between stimuli). However, Kellicut-Jones and Sellers [15] found no differences between the two stimulus sizes when a checkerboard flash pattern was employed. Reichert et al. [16] used a paradigm that was not based on a matrix but involved only two lateralized stimuli (they relied on the lateralized potential N2pc) and also did not find a significant effect related to stimulus size. Therefore, it seems that the effect of the stimulus size may differ according to the type of ERP-based paradigm used to control a BCI.

RSVP is a gaze-independent control paradigm widely used in the field of BCIs. Therefore, the study of the effect of stimulus size on system performance could be a significant contribution to improving the performance of its potential users. Therefore, the present study will evaluate the effect of stimulus size on performance in an ERP-BCI under RSVP.

## 2. Materials and Methods

### 2.1. Participants

Our experiment involved 12 participants (aged 23.58 ± 2.43, 4 female) who had a normal or corrected-to-normal vision. None of the participants had previous experience controlling BCI systems. The study was approved by the Ethics Committee of the University of Malaga and met the ethical standards of the Helsinki Declaration. According to self-reports, none of the participants had any history of neurological or psychiatric illness, and none were taking any medication regularly. All participants provided written consent.

### 2.2. Data Acquisition and Signal Processing

EEG data were recorded at a sample rate of 250 Hz using electrode positions Fz, Cz, Pz, Oz, P3, P4, PO7, and PO8, according to the 10/10 international system. All channels were referenced to the left mastoid and grounded to position AFz. Impedance was reduced to below 10 kΩ for each electrode with electrolyte gel. Signals were amplified by an acti-Champ amplifier (Brain Products GmbH, Gilching, Germany), and a band-pass filter at 0.1–30 Hz was used. The BCI software tool used to display the graphical interface to participants and collect data was BCI2000 [17]. Once all the data had been registered and the session had ended, artifacts in the data were corrected through the artifact subspace reconstruction (ASR) algorithm using the default settings in EEGLAB and the Riemannian distance [18,19]. Then, a stepwise linear discriminant analysis of the data was performed to obtain the weights of the classifier and to calculate the accuracy (using the BCI2000 tool called P300Classifier).

### 2.3. Experimental Conditions

We explored three conditions based on the size of the stimuli. The three conditions were named according to the size of the stimuli used as small, medium, and large (see Table 1 for the specific measures for each condition). The small size was chosen as the minimum possible, although an attempt was made to ensure that it could be correctly visualized by a non-vision-impaired user. The user’s ability to discriminate between stimuli was checked during the session (see Section 2.4.2). The variable that was manipulated to control the font size in BCI2000 was CaptionHeight, which was multiplied by 11 for each size increase. The number of different stimuli used for each condition was 10. The stimuli used for the conditions were letters (A, C, D, E, H, I, N, O, P, and R). Initially, the letters were simply discriminative stimuli using small sizes, which could be made harder through the use of more complex stimuli (e.g., photographs). It was also convenient to use letters, as they are the most commonly used stimuli in BCI systems for the development of spellers [5].

The size of the screen was 46.47 × 31.08 cm (16:10 ratio, 22 in, 55.9 cm, Acer P224W connected via HDMI), with a resolution of 1680 × 1050 pixels. The refresh rate of the screen was 60.014 Hz. The color used for the letters was black (#000000), and white (#FFFFFF) was used for the background. The font used for the letters displayed by BCI2000 was Arial bold. The distance between the participant’s viewing point and the screen was set to 60 cm.

Regarding the speed of stimuli presentation, a stimulus onset asynchrony (SOA) of 288 ms was used, and an inter-stimulus interval (ISI) of 160 ms was applied such that each stimulus was presented for 128 ms.

### 2.4. Procedure

When each participant arrived at the laboratory, the test was explained to them, they signed the informed consent form, and the instrumentation was prepared. The experiment took place in a single session divided into three parts: (i) a BCI task; (ii) two perceptual tasks, one just before the BCI task (the pre-BCI task, carried out once the instrumentation had been prepared) and the other after it (post-BCI task); and (iii) an item regarding the subjective difficulty of completing the task with each size. The purpose of each part of the session was as follows. The aim of the BCI task was to obtain the accuracy of the system with each stimulus size in order to establish a comparison between them, and there was no online feedback in which the user actually controlled the interface. The purpose of the perceptual task was to evaluate the user’s competence in terms of perceiving each of the stimuli in different sizes. Finally, the subjective difficulty item was used to obtain a satisfaction measure to assess the usability of each size. The three stimulus size conditions were tested for both the BCI task and the perceptual tasks. An intra-subject (also known as repeated measures) design was used, meaning that all users tested all three conditions. The tasks involved in the experiment are described in the following.

#### 2.4.1. Brain–Computer Interface Task

In this task, the user was asked to focus his/her attention on one of the letters indicated (i.e., the target letter). The objective was for the BCI system to be able to discriminate between the ERP signal associated with the target stimulus and the non-target ones. The user did not receive online feedback since it was an exclusively offline BCI task, and their performance was calculated once the session had ended. The process of selecting a letter in the interface was referred to as a trial and involved the action of selecting a target letter from the 10 possible letter options shown in the interface. During a trial, each letter was presented five times. The presentation of each of the letters available on the interface corresponded to a sequence, and hence a trial was composed of five sequences. The order of presentation of the letters in each sequence was random, without replacement. The user was asked to mentally count the number of presentations of the target letter in order to ensure that his/her attention was focused on the task. Before starting each trial, there was a pause of 3840 ms. At the beginning of this pause, the letter to be attended to in the next trial was indicated by the message “*Atiende a*” (“Focus on”, in Spanish, 960 ms), and the target letter was then immediately presented in the same position (128 ms). The first stimulus of the trial was presented 2752 ms later. When all the trial sequences had finished, there was another pause of 1920 ms. Hence, between trials, there was a total pause of 5760 ms (3840 ms pre-trial and 1920 ms post-trial).

The duration of a single trial was 14,240 ms: 50 stimulus presentations (five sequences × 10 letters) of 128 ms and 49 inter-stimulus intervals (ISI) of 160 ms. A block was also defined as the set of trials from when the application was started by the experimenter until it was automatically stopped. In the present experiment, each block consisted of 10 selections (trials), in which 10 letters were focused on in order to complete two Spanish words of five letters each, without a space between them. For each condition, 3 blocks were carried out (30 selections per condition, 3 blocks × 10 trials). The first block was “NICHOPARED” (*nicho* and *pared*, niche and wall in Spanish); the second block was “CHINOPADRE” (*chino* and *padre*, Chinese and father in Spanish); and the third block was “HINCOPREDA” (*hinco* and *preda*, sink and booty in Spanish) (Figure 1). These words were chosen in such a way that in each block, all the available letters were used for a trial only once, and the words were never repeated between the different blocks under the same conditions. The participants were not presented with three blocks of the same size consecutively; instead, they tested three blocks with different sizes for the first pair of words (“NICHOPARED”) and then tested the three sizes again for the second pair of words (“CHINOPADRE”). Finally, they completed the third block with each size using the last pair of words (“HINCOPREDA”). For each pair of words, the order of the conditions for a specific participant was different (e.g., for participant P01, for “NICHOPARED”, the order was small, medium, and then large; for “CHINOPADRE”, the order was large, small, and then medium; and for “HINCOPREDA”, the order was medium, large and then small). The order was also counterbalanced—i.e., evenly distributed—across participants to prevent any unwanted effects such as learning or fatigue, and all conditions were equally distributed.

#### 2.4.2. Perceptual Task and Subjective Difficulty

The perceptual verification task was carried out twice (two blocks) for each condition: once before the BCI task and once after. The task consisted of a test using the same presentation paradigm as in the BCI task, in which each stimulus was presented between three and seven times (this number was fixed to five for each trial in the BCI task). The total number of stimulus presentations during a trial was 50, in the same way as the BCI task. After each trial, the user was tasked to write down on paper the number of times he/she had seen the target letter. The time available to the user to write his/her answers after each trial was 4800 ms (that is, the post-trial pause, which in the BCI task was equal to 1920 ms). The rest of the temporal parameters were the same as in the BCI task. In a similar way to the BCI task, 10 letters needed to be attended to in each block. The order of the letters that the user had to count in each block for the three conditions was always the same: “NICHOPARED” (10 letters to count, giving 10 trials).

There were three possible templates for each block (named A, B, and C), which varied in terms of the number of times each letter appeared (Table 2). The goal of using these different templates was that the number of times each letter appeared in each trial and condition was different. In the first verification (pre-BCI) task the order of the templates was always A, B, and C, while for the second (post-BCI) task, it was C, A, and B. The order of the conditions (sizes) was the same for the first and second verification tasks for all participants (e.g., if P01′s order was small, medium, and then large for the first verification task, the same order was used for his/her second task). However, the order of conditions was counterbalanced between participants so that each participant had a different order. For example, for S01, the orders of the first and second verification tasks and templates were small A, medium B, and large C, and then small C, medium A, and large B, while for S02, they were large A, small B, and medium C, and then large C, small A, and medium B. The use of these templates was unknown to the participant, and they were instructed to assume that the number of presentations for each letter was random.

Finally, after the last block of the perceptual task, the participant had to answer the item related to the subjective difficulty of performing the BCI task with each of the different sizes. Once these items had been answered, the session was over.

### 2.5. Evaluation

In order to explore the effect of the stimulus size factor, four dependent variables were analyzed: (i) the accuracy of the BCI task; (ii) the amplitude (µV) of the ERP waveform during the BCI task; (iii) the errors in the perceptual task; and (iv) a subjective item regarding fatigue.

*Accuracy on the BCI task*. The system classification accuracy (i.e., the number of correctly predicted selections divided by the total number of predicted selections) was obtained by applying a three-fold cross-validation method to assess the performance for the different conditions. The analysis consisted of a comparison between the accuracies obtained for the different size conditions (small, medium, and large) and sequences (from one to five).

*Amplitude of the ERP waveform in the BCI task*. The amplitude (µV) of the ERP waveform (from −200 to 800 ms) was studied in order to observe how the different stimulus sizes affected this variable for target and non-target stimuli. Two analyses were performed: (i) a comparison between conditions as a function of stimulus type (i.e., target and non-target stimulus) for each channel; and (ii) a comparison between conditions using as a variable the amplitude difference (µV) between the target and the non-target amplitudes for each channel. To perform these analyses, the electrode signal was corrected using an interval period from −200 to 0 ms as a baseline.

*Errors in the perceptual task*. For the perceptual task, the difference in absolute value between the number counted by the participant and the correct number of times each letter was presented was considered the dependent variable. That is, if a participant counted five when the actual number of presentations for that letter was seven, this was counted as two error points. For each block, the number of letter presentations was equal to 50, meaning that the error variable in the perceptual task could take values of between 0 and 50 for each block.

*Subjective difficulty*. Finally, for each condition, a subjective item was used to measure the level of difficulty felt by the user (range 0–10, integers) to carry out the required tasks. The item used was the following: “How difficult was it for you to perform the tasks using the [for example] small condition?”.

The software packages used for data analysis were JASP for accuracy during the BCI task, the errors in the perceptual tasks, and the score in the subjective difficulty [20], and EEGLAB for the amplitude of the ERP waveform in the BCI task [19]. The analyses carried out with JASP were different repeated measures ANOVAs. The purpose of this test was to evaluate whether a given factor affected the scores obtained on a certain variable (e.g., stimulus size on accuracy). If a factor showed a significant effect (*p*-value below 0.05), multiple comparison analyses were performed using Student’s *t*-test for repeated measures. These tests allowed us to know from which specific factor levels (e.g., large, medium, and small sizes) significant differences in the variable (e.g., accuracy) were found (again, a *p*-value below 0.05). For these factors analyzed with JASP, Greenhouse–Geisser corrections were used for violation of the sphericity assumption, and eta-squared (*η*^2^) was used as a measure of effect size. In multiple comparisons, the computation of Cohen’s *d* effect size was based on pooled error. On the other hand, in the analyses carried out with EEGLAB, although repeated measures ANOVAs were also performed, no multiple comparisons were made for specific differences between sizes. Additionally, in all the multiple comparisons carried out, both with JASP and EEGLAB, the Holm correction method was applied to calculate the *p*-value. As indicated above, the *p*-value threshold at which the null hypothesis (H_0_, no differences between conditions) will be accepted is equal to 0.05. Any value below this threshold will allow us to declare that the probability of committing a type I error—rejecting the null hypothesis when it is true—appears small enough, and therefore, the alternative hypothesis (H_1_, there are differences between conditions) will be accepted.

## 3. Results

### 3.1. Brain–Computer Interface Task

For the BCI accuracy, a two-way 3 × 5 repeated measure ANOVA (three sizes, five sequences) showed a significant effect for stimulus size (*F* = 21.677; *p* < 0.001; *η*^2^ = 0.17) and sequence (*F* = 154.243; *p* < 0.001; *η*^2^ = 0.623), as well as an interaction effect between both factors (*F* = 2.281; *p* = 0.029; *η*^2^ = 0.012) (Figure 2). It can therefore be stated that accuracy is affected by both the size of the stimuli and the number of sequences presented. Regarding the size factor, specific differences were found between all sizes: small versus medium (*p* = 0.03; *d* = −0.565), small versus large (*p* < 0.001; *d* = −1.215), and medium versus large (*p* = 0.032; *d* = −0.65). For the factor relating to sequences, significant differences were found between all possible pairs: sequence 1 versus sequence 2 (*p* < 0.001; *d* = −1.296), sequence 3 (*p* < 0.001; *d* = −1.959), sequence 4 (*p* < 0.001; *d* = −2.416) and sequence 5 (*p* < 0.001; *d* = −2.681); sequence 2 versus sequence 3 (*p* < 0.001; *d* = −0.663), sequence 4 (*p* < 0.001; *d* = −1.119) and sequence 5 (*p* < 0.001; *d* = −1.385); sequence 3 versus sequence 4 (*p* < 0.001; *d* = −0.456) and sequence 5 (*p* < 0.001; *d* = −0.722); and sequence 4 versus sequence 5 (*p* = 0.002; *d* = −0.266). Finally, due to the interaction effect found between both factors (size × sequence), a post hoc analysis of the different sizes in each of the sequences was carried out, as shown in Table 3. These results indicate that the specific differences between conditions depend on the number of sequences presented.

### 3.2. Perceptual Task and Subjective Difficulty

In reference to the perceptual analysis, a two-way 3 × 2 ANOVA (three sizes, two times) was performed. This analysis showed no significant effect for the size factor (*F* = 0.209; *p* = 0.813; *η*^2^ = 0.008), for time (*F* = 3.555; *p* = 0.086; *η*^2^ = 0.107), or for the size × time interaction (*F* = 0.888; *p* = 0.426; *η*^2^ = 0.011) (Figure 3). It cannot therefore be claimed that the stimulus size (small, medium, and large) or the timing of the test (pre-BCI and post-BCI tasks) influenced the number of errors reported in the perceptual task.

For the item related to subjective difficulty, a one-way 3 ANOVA (three sizes) was carried out, which showed a significant main effect for size (*F* = 23.271; *p* < 0.001; *η*^2^ = 0.679) (Figure 4). The mean values for each of the sizes were as follows: small: 6.42 ± 1.78 points; medium: 3.17 ± 2.04 points; large: 3.25 ± 1.82 points. Specifically, significant differences were found for small versus medium (*p* < 0.001; *d* = 1.727) and large (*p* < 0.001; *d* = 1.683) but not for medium versus large (*p* = 1; *d* = −0.044). The results clearly indicate that the small size was reported to be the most difficult for the participants in the BCI task.

### 3.3. Event-Related Potential Waveform

Figure 5 shows the ERP waveform and the results of a statistical analysis comparing the amplitude of each size for the target and non-target stimuli and the amplitude differences between them. In most channels, significant differences between sizes were found for both target and non-target stimuli. For both types of stimuli, these changes seem to be led by potentials with higher amplitude (either positive or negative) for the large size compared to the other sizes. These oscillations are probably visually evoked potentials (VEPs), i.e., changes produced by the periodic presentation of any visual stimulus (target or non-target). Since the signals for both stimuli seem to be affected by the presence of VEPs, it is convenient to calculate the amplitude difference, as this should eliminate the effects shared by both types of stimuli and highlight only the specific components produced by the presentation of a stimulus that the participants are expecting. It is also likely that the amplitude difference is a useful variable for analyzing classifier performance since the greater the difference between the signals associated with the target and non-target stimuli, the easier it will be for the classifier to perform its task.

The last two rows of Figure 5 show the results for the amplitude difference variable for each of the conditions and each of the EEG channels. Two types of significant intervals were found: significant differences were found in the Fz and Cz channels (around 410–460 ms, depending on the channel), which were possibly related to the ERP P3 component, while for the channels recorded in the occipital region (PO7, PO8, and Oz) a significant interval produced by negativity (around 340–400 ms, depending on the channel), possibly N2, was found.

## 4. Discussion

The aim of the present work was to improve the performance of an ERP-BCI using a gaze-independent control paradigm. Therefore, we proceeded to study whether stimulus size could influence the performance of an ERP-BCI under RSVP. Indeed, the results presented in the previous section indicate that the size of the stimuli used in an ERP-BCI under RSVP influences user performance. In addition, the users’ ability to discriminate between stimuli, the subjective difficulty involved in using each size, and the EEG signal recorded were evaluated. In the present section, these results are discussed and contextualized in regard to related literature.

### 4.1. Brain–Computer Interface Task

The studies by Li et al. [21] and Ron-Angevin et al. [14] involved an RCP-based presentation mode and showed that the size of the matrix (including the stimulus size and distance between rows and columns) had an impact on BCI performance; however, the conclusions of these studies were opposed to each other. Ron-Angevin et al. [14] observed that the best performing size was the small one (0.4 × 0.4 degrees approx. of stimulus size), while Li et al. [21] found that displaying the matrix on a computer monitor (17 in screen, matrix and stimulus size unspecified) gave the best performance in comparison to a global positioning system monitor (9 in screen) and a cell phone (5 in screen). The results of the present work with RSVP are aligned with those of Li et al. [21]: a larger size of the stimulus leads to better performance of the ERP-BCI. However, it is important to highlight that the differences between sizes in the present study varied depending on the sequence, as indicated by the size × sequence interaction effect. For example, if the ERP-BCI under RSVP had been using one sequence (e.g., when a BCI is used to detect the occurrence of the desired event that the user is watching for, such as the appearance of an airplane on a radar), no difference would have been noted between the small and medium sizes (39.72% versus 42.22%, respectively), but if the application had been using five sequences (e.g., in the case of a speller or any other application where the number of sequences can be manipulated), it is important to consider that a significant difference would have been found between small and medium sizes (75.14% versus 86.39%, respectively). Hence, the specific differences between sizes depending on how many sequences are used. In the present study, the main trend was very clear: the larger the stimulus size, the better the performance.

### 4.2. Perceptual Task and Subjective Difficulty

The results of the perceptual analysis show that there were no significant differences between either the different sizes (size factor) or the pre- and post-BCI task measures (time factor). The results for the size factor are relevant because they indicate that the participants recognized the stimuli in a similar way, i.e., regardless of the size used. In addition, the hypothesis that the performance decrease for smaller sizes (small versus medium and large, and medium versus large) was due to poor perception of the stimuli can be rejected, as significant differences in performance were also shown between the medium and large conditions. Indeed, the differences between the medium and large sizes cannot be explained by incorrect recognition of medium stimuli due to their size (the medium stimulus size was around 1.81 × 1.72 degrees, 1.9 × 1.8 cm at a distance of 60 cm, i.e., easily recognizable by a user with normal or corrected-to-normal vision). With regard to the time factor, although there was no significant difference, the tendency towards an increase in errors from the pre-BCI task to the post-BCI task should be noted. This could be due to a detrimental effect related to the time variable since the session (without feedback, which could be quite boring) lasted approximately 80 min, and many of the participants indicated that they ended up quite tired, sleepy, or even with some visual and/or mental fatigue, which has been found to be a problem in previous ERP studies [22].

Finally, in reference to the subjective difficulty item, it was clear that the small size condition was the worst rated by the users. If these last two results are considered together (although the small size was associated with greater difficulty, there were no significant differences in the number of errors in the perceptual task), it is possible to assume that the users may expend additional effort to reach the performance obtained using the small size condition. Even if the user demonstrates good visual performance in recognition of stimuli (and in the BCI task), the need for this additional effort may pose a problem when the user’s comfort is prioritized or in longer sessions.

### 4.3. Event-Related Potential Waveform

The goal of the classifier is to discriminate as accurately as possible between the signals corresponding to the target and non-target stimuli. Previous studies have found that the higher the amplitude difference, the better the BCI performance (e.g., [13,23]. Considering the results obtained in Figure 5, it could be inferred that the better performance under the conditions with a larger size is likely to be related to the ERP components where significant differences have been found: the P3 component in Fz and Cz and the N2 component in PO7, PO8, and Oz. These components have previously been reported as the two most remarkable potentials in an ERP-BCI under RSVP [7], and it therefore makes sense that this increased amplitude difference is related to an improvement in performance.

### 4.4. Limitations

The present work has certain limitations that should be considered, either to clarify the impact of our findings or to be resolved in future studies. The potential limitations of the present work that will be addressed are (i) the young and healthy target population; (ii) the lack of an online task; and (iii) the use of excessively different sizes. In addition to detailing these limitations, we will explain why these decisions were made.

Firstly, as described in previous works, the need to validate results in the clinical population remains a challenge in the BCI field [24]. In the present study, most of the participants were young adult college students with no reports of neurological problems. This population differed substantially from the main target populations of ERP-BCI under RSVP, who are middle-aged or elderly patients with severe motor problems. Caution is therefore required in generalizing the results obtained here to that clinical population without performing corresponding tests. However, not all BCI applications are intended for patients; some are aimed at a non-clinical population, and these could benefit from the findings presented here regarding the stimulus size [25].

Secondly, in the present work, users only participated in an offline (calibration) task and not in an online task, meaning that they did not really control the proposed system or receive real-time feedback on their performance. This online control is necessary to meet the goal of providing a useful tool for users to interact with their environment. However, the reliability of the results was validated through a cross-validation method to ensure that even if the users did not receive online feedback, the system could be guaranteed to be working properly. Avoiding the use of an online task also allowed for a more extensive offline task and hence more reliable results (as more trials are performed, the average accuracies and ERP signals of the participants become more reliable).

Finally, the sizes selected for the comparison were very different. Each size was 11 times larger than the next smaller one (the BCI2000 CaptionHeight variable was equal to 0.8 for small, 8.8 for medium, and 96.8 for large). It is therefore possible that the impact of the size factor would not be so significant if two slightly different sizes were compared. As an initial approach to this issue, we advocate starting with the options where it is easiest to find an effect. Likewise, the results of the perceptual task also indicated that the size of the stimuli did not affect the users’ ability to perceive them. It would also be interesting to explore whether it was possible to achieve BCI control with such small stimuli (around 0.1 × 0.1 degrees, 0.1 × 0.1 cm at a distance of 60 cm) to consider the use of devices that allowed for the largest possible stimuli size, or to assess the use of other types of presentation strategies that allow for maximization of their size (for example, augmented reality [26]).

## 5. Conclusions

To our knowledge, this is the first work on an ERP-BCI under RSVP that has evaluated the effect of three stimulus sizes. Our results indicated that size has a significant effect on performance (H_1_ is accepted); specifically, in every comparison between stimulus sizes, the condition with the larger stimulus size performed better than the smaller one (large > medium > small). Although users reported greater difficulty on the task with the smallest stimuli (H_1_ is accepted), the effect on performance did not appear to be explained by incorrect detection, as all stimuli were recognized similarly on the perceptual task (H_0_ is accepted).

The findings obtained in the present work may be useful for the design of future ERP-BCIs that need to be controlled by users without oculomotor control. However, due to the limitations previously mentioned (Section 4.4), it is important to consolidate the knowledge acquired in order that this type of technology can eventually help its potential users. Therefore, the verification of previously obtained findings is strongly recommended to consolidate them and build a solid base. In addition, our findings regarding stimulus size may raise new questions for future studies. Some examples of possible studies are as follows. Firstly, since the use of large screens could lead to practical and/or spatial issues in the user’s room, it would be interesting to apply virtual or augmented reality techniques to maximize the size of the stimuli (e.g., Kim et al. (2021) [27]). Secondly, in the present work, we used letters because these have simple traces and are easier to discriminate at tiny sizes. However, previous studies have shown that the type of stimulus used has an impact on the performance of an ERP-BCI under RSVP (e.g., Chen et al. (2016) [10] and Lees et al. (2020) [11]). It would therefore be interesting to test the impact of stimulus size on performance using alternative stimuli to control an ERP-BCI under RSVP. Thirdly, a previous study showed that under overt attention, the level of overlap between stimuli is also a variable that negatively influences performance [28]. Hence, in an area-limited interface, it would be necessary to choose what is more convenient: presenting large but overlapping stimuli or small but non-overlapping/far apart stimuli. It might also be interesting to carry out a study to compare both factors, considering also the possible intermediate options). Finally, as mentioned in the discussion, it is important to evaluate the results presented here on the target population for these applications (patients with severe motor problems) in an online task.

In short, the present work showed that stimulus size was a factor that was positively related to users’ performance in terms of controlling an ERP-BCI under RSVP. We have also described how this finding opens the door to further research studies and proposals.

## Figures and Tables

**Figure 1 sensors-22-09505-f001:**
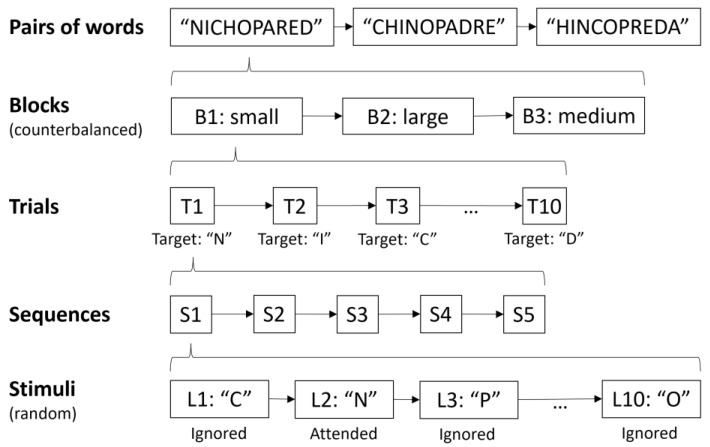
Experimental procedure followed by the participants. The order of the conditions (small, medium, and large) used for each pair of words was counterbalanced for the same participant (i.e., each pair of words had a different order) and between different participants. Likewise, the order of presentation of the stimuli in each sequence was random, without replacement.

**Figure 2 sensors-22-09505-f002:**
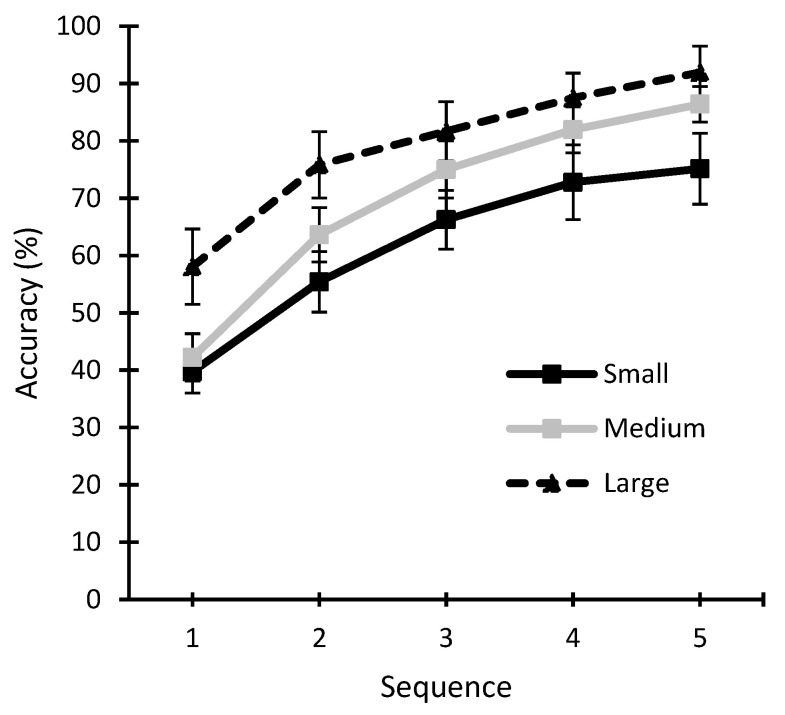
Accuracy (mean ± standard error) of the different conditions (small, medium, and large sizes) as a function of the number of sequences used.

**Figure 3 sensors-22-09505-f003:**
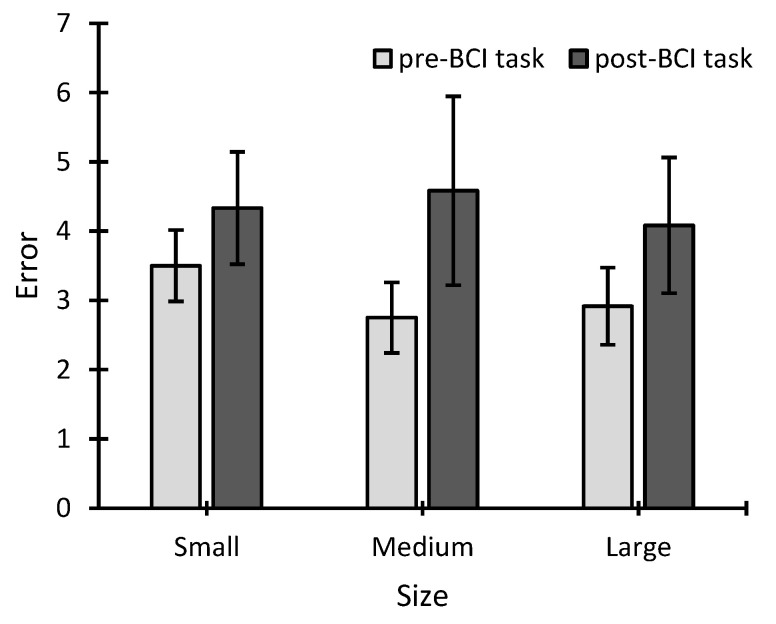
Errors (mean ± standard error, for 50 stimulus presentations in each column) during the perceptual task, for each size, for both the perceptual tasks before and after the brain–computer interface task.

**Figure 4 sensors-22-09505-f004:**
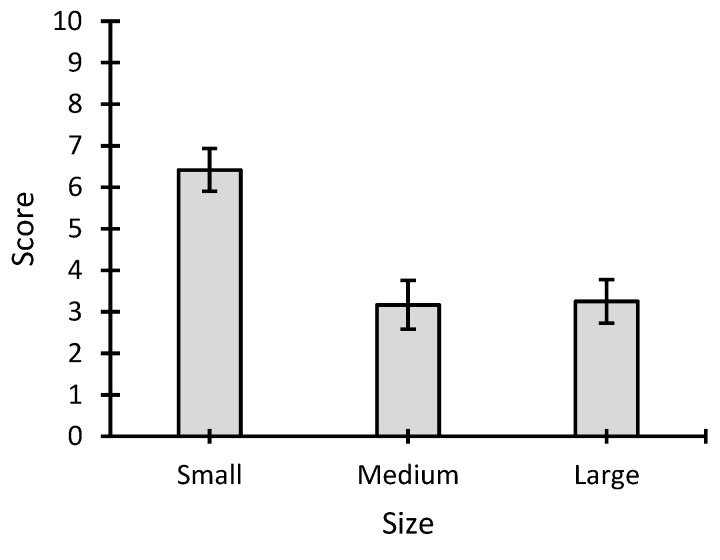
Score (mean ± standard error) for the item related to the subjective difficulty in performing the brain–computer interface task for each stimulus size (small, medium, and large).

**Figure 5 sensors-22-09505-f005:**
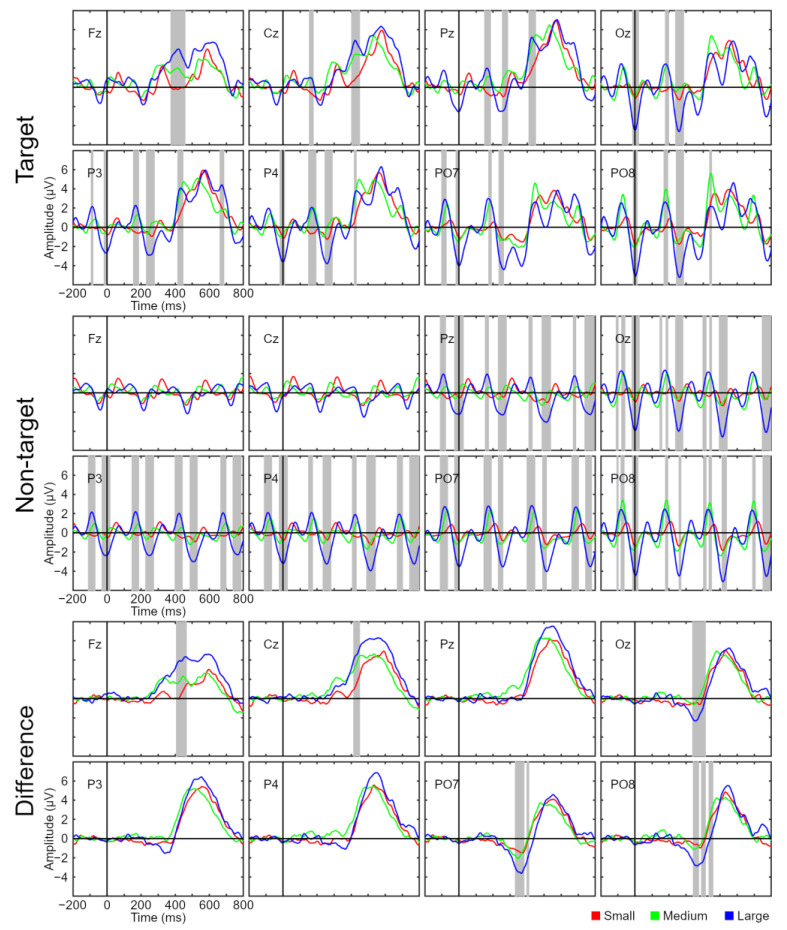
Grand average event-related potential waveforms for the target and non-target stimuli signals and the amplitude differences between them for all channels used (Fz, Cz, Pz, Oz, P3, P4, PO7, and PO8) and for the three conditions (small, medium, and large). Significant intervals are denoted with a gray background on the time axis. The Holm correction method was applied.

**Table 1 sensors-22-09505-t001:** Values of the stimulus size parameters.

Condition	Size		
	Degrees ^1^	Centimeters ^1^	Caption Height ^2^
Small	0.1 × 0.1	0.1 × 0.1	0.8
Medium	1.72 × 1.81	1.8 × 1.9	8.8
Large	27.98 × 29.15	19.9 × 20.05	96.8

^1^ For these measurements, i.e., the visual angle in degrees and length in centimeters, the letter O was used as a reference, and the dimensions are reported as width × height. ^2^ This is a BCI2000-specific parameter that indicates the height of the stimulus caption text as a percentage of screen height. Note: The participant’s viewing point and the screen were separated by 60 cm.

**Table 2 sensors-22-09505-t002:** Number of presentations for each letter according to the template used.

Template	Letters to be Focused On									
	N	I	C	H	O	P	A	R	E	D
A	7	3	6	5	4	7	4	6	3	5
B	5	6	3	7	3	6	5	4	7	4
C	4	4	7	4	6	3	7	3	6	6

**Table 3 sensors-22-09505-t003:** Accuracy (%, mean ± standard deviation) comparisons between the four different spellers for sequences 1–5.

Sequence	Size		
	(1) Small	(2) Medium	(3) Large
1	39.72 ± 11.05 ^3^	42.22 ± 11.13 ^3^	58.06 ± 15.92 ^1,2^
2	55.42 ± 15.43 ^2,3^	63.61 ± 16.05 ^1,3^	75.83 ± 13.19 ^1,2^
3	66.25 ± 15.18 ^3^	75 ± 15.34	81.67 ± 13.67 ^1^
4	72.78 ± 16.2 ^2,3^	81.94 ± 12.35 ^1^	87.5 ± 14.57 ^1^
5	75.14 ± 18.05 ^2,3^	86.39 ± 9.69 ^1,3^	91.95 ± 10.77 ^1,2^

Note: significant differences between conditions (*p* < 0.05) are denoted with a superindex to show which condition average they are different to (1 for small, 2 for medium, and 3 for large). The Holm correction method for multiple comparisons was applied.

## Data Availability

The data presented in this study are available on request from the corresponding author.

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
