# Peer review of "Effect of Stimulus Size in a Visual ERP-Based BCI under RSVP"

_sensors, 2022, doi:10.3390/s22239505_

Round 1

Reviewer 1 Report

 The This paper shows that stimulus size is a relevant factor in the performance of an ERP-BCI under RSVP. The comments are given as following :

1. There are 11 experimental samples only in this study, however there are three limitations for applications. From the practical view point. the authors should describe more reasonable statements for contribution of this work.

2.No scale units are showed for non-target and target cases in Fig. 5, respectively

3. The results indicated that size has a significant effect on  performance; specifically, the larger the stimulus size, the higher the user performance.  This results are too rough for academic expression. It has to be described by exact formula or compact expression.

Reviewer 2 Report

The presented work is to assess whether stimulus size has an impact on ERP-BCI performance under the RSVP paradigm. During the performance test twelve participants tested the ERP-BCI under RSVP using three different stimulus sizes: small, medium, and large. The presented results showed significant differences in accuracy between the conditions; the larger the stimulus, the better the accuracy obtained. The authors shown that these differences were not due to incorrect perception of the stimuli, since there was no effect from size in a perceptual discrimination task. The paper clearly shows that stimulus size is a relevant factor in the performance of an ERP-BCI under RSVP.

The abstract is a good description of the work. The introduction is well structured, and it covers most of the concepts investigated in the methodological part. In the introduction section, the research goals and the research subject and research questions should be defined more precisely. The introduction does not specify properly the contributions of the paper. The author must explain how his work is different than other similar papers. Research questions must be explained in more details. Abstract must focus only on the problem and will the paper will help in solving this problem. Please, clearly identify the contributions of the study. Please explain exactly what impact does this research have? I think some HCI references should be added to clearly classify the topic of the paper in the field of human computer iinteraction. Besides the mentioned research papers there are several other systems like BCIs, eye-tracking methods are applied nowadays and some cognitive aspects and research is relevant to this field. It would be good to see some sentences introducing the wide variety of applications of human-computer feedback systems like Examine the effect of different web-based media on human brainwaves; Study of algorithmic problem-solving and executive function; Assessing visual attention in children using gp3 eye tracker; EEG-based computer control Interface for brain-machine interaction and The analysis of hand gesture based cursor position control during solve an IT related task.  Several other human-computer based monitoring systems is used in this field so please summarize these methods and applications in the introduction like quantitative analysis of relationship between visual attention and eye-hand coordination; control of incoming calls by a windows phone based brain computer interface; electroencephalogram-based brain-computer interface for internet of robotic things and evaluation of eye-movement metrics in a software debugging task using gp3 eye tracker. 

In the discussion section, these goals and research proposals should be clearly responded to in the light of the results obtained. How were you convinced of the validity and reliabilityof the system? Please describe the method of the evaluation considering the validity and reliability of the system. 

It is essential to make up for the shortcomings raised above for the proper discussion of the results of the paper and its placement in the field of HCI.

Round 2

Reviewer 1 Report

The authors' reply to this comment, "(1.3) The results indicated that size has a significant effect on performance; specifically, the larger the stimulus size, the higher the user performance. This results are too rough for academic expression. It has to be described by exact formula or compact expression.", should be more accurate description in mathmatic sense. 

Reviewer 2 Report

Acceptable

Author Response

Thank you very much for all your previous comments and help us to improve the quality of the paper. We are pleased to read that you consider the work acceptable for publication.

Round 3

Reviewer 1 Report

No more comments.